# The value of confidence: Confidence prediction errors drive value-based learning in the absence of external feedback

**Lena Esther Ptasczynski**[1,2]*, **Isa Steinecker**[1,3], **Philipp Sterzer**[1,2,3,4], **Matthias Guggenmos**[1,5]

**1** Charité—Universitätsmedizin Berlin, corporate member of Freie Universität Berlin and Humboldt-Universität zu Berlin, Department of Psychiatry and Neurosciences, Berlin, Germany, **2** Berlin School of Mind and Brain, Humboldt-Universität zu Berlin, Berlin, Germany, **3** Bernstein Center for Computational Neuroscience, corporate member of Humboldt-Universität zu Berlin, Berlin, Germany, **4** Universitäre Psychiatrische Kliniken Basel, University of Basel, Basel, Switzerland, **5** Health and Medical University, Institute for Mind, Brain and Behavior, Potsdam, Germany

* lena-esther.ptasczynski@charite.de

**Data Availability Statement:** The experimental data, analysis scripts as well as computational models for the current study are available through the ConfLearning GitHub repository: https://github.

## Abstract

Reinforcement learning algorithms have a long-standing success story in explaining the dynamics of instrumental conditioning in humans and other species. While normative reinforcement learning models are critically dependent on external feedback, recent findings in the field of perceptual learning point to a crucial role of internally generated reinforcement signals based on subjective confidence, when external feedback is not available. Here, we investigated the existence of such confidence-based learning signals in a key domain of reinforcement-based learning: instrumental conditioning. We conducted a value-based decision making experiment which included phases with and without external feedback and in which participants reported their confidence in addition to choices. Behaviorally, we found signatures of self-reinforcement in phases without feedback, reflected in an increase of subjective confidence and choice consistency. To clarify the mechanistic role of confidence in value-based learning, we compared a family of confidence-based learning models with more standard models predicting either no change in value estimates or a devaluation over time when no external reward is provided. We found that confidence-based models indeed outperformed these reference models, whereby the learning signal of the winning model was based on the prediction error between current confidence and a stimulus-unspecific average of previous confidence levels. Interestingly, individuals with more volatile reward-based value updates in the presence of feedback also showed more volatile confidence-based value updates when feedback was not available. Together, our results provide evidence that confidence-based learning signals affect instrumentally learned subjective values in the absence of external feedback.

com/eptas/ConfLearning. We used Zenodo to assign a DOI to this repository (DOI 10.5281/zenodo.7148520).

**Funding:** This study was supported by the grants GU 1845/1-1 and STE 1430/9-1 to MG and PS from the German Research Foundation (DFG; www.dfg.de), a Clinical Fellowship to PS from the Berlin Institute of Health (www.bihealth.org) and a Mind & Brain scholarship to LEP from the Berlin School of Mind and Brain, Humboldt-Universität zu Berlin (www.mind-and-brain.de). The funders had no role in study design, data collection and analysis, decision to publish, or preparation of the manuscript.

**Competing interests:** The authors have declared that no competing interests exist.

## Author summary

Reinforcement learning models successfully simulate value-based learning processes (e.g., "How worthwhile is it to choose the same option again?") when external reward feedback is provided (e.g., drops of sweet liquids or money). But does learning stagnate if such feedback is no longer provided? Recently, a number of studies have shown that subjective confidence can likewise act as an internal reward signal, when external feedback is not available. These results are in line with the intuitive experience that being confident about choices and actions comes with a satisfying feeling of accomplishment. To better understand the role of confidence in value-based learning, we designed a study in which participants had to learn the value of choice options in phases with and without external feedback. Behaviorally, we found signatures of self-reinforcement, such as increased confidence and choice consistency, in phases without feedback. To examine the underlying mechanisms, we compared computational models, in which learning was guided by confidence signals, with more standard reinforcement learning models. A statistical comparison of these models showed that a confidence-based model in which generic confidence prediction errors (e.g., "Am I as confident as expected?") guide learning indeed outperformed the standard models.

## Introduction

The reinforcement learning principle, according to which learning is controlled by action-contingent feedback, explains fundamental forms of learning across many modalities and species [1]. Yet, there are important instances of learning that occur in the absence of external feedback, and which thus challenge the generality of this model class.

A prominent example is perceptual learning, for which behavioral improvements are frequently found through training or mere exposure and without any external feedback [2–6]. Moreover, the (subjective) sense of accomplishment in an unrelated task likewise induces perceptual learning, even in the absence of stimulus awareness [7,8]. Together, these findings have led to the notion of a 'diffuse internal reward signal' [9], i.e. a reinforcement signal that is triggered based on some form of internal feedback.

More recently, such internal feedback signals have been investigated by means of fMRI, operationalized in the form of confidence reports [10–13]. The consistent finding of these studies was that confidence-based learning signals engaged a network of brain regions that has previously been identified for the coding of *reward* prediction errors [14], including the ventral striatum (a dopaminergic target region) and the ventral tegmental area (a dopaminergic source region). In line with these neurobiological observations, a recent study has shown that having confidence in one's own actions is associated with a feeling of increased pleasantness and satisfaction [15]. Together, these findings suggest that learning based on external and internal feedback operates on a shared neural mechanism.

In the present study, we aimed to examine the generality of such putative confidence-based learning signals. We hypothesized that, if confidence in actions indeed takes the form of a diffuse internal reward signal, it may also affect the subjective values of these actions, similar to instances of external reinforcement. What could be the benefit of such a self-reinforcement mechanism in the context of value-based decision making? Although we prefer to be agnostic at this point about whether such a mechanism would be adaptive, self-reinforcement could strengthen previously learned preferences to make them more robust in the face of decision noise and potential memory leakage. On the other hand it is possible that such a mechanism is

a general automatic concomitant of learning and decision making without external feedback that, while beneficial in some domains, may not be useful or even maladaptive in others, including value-based decision making.

Indeed, the notion that subjective values change in the absence of external feedback – without an obvious adaptive benefit – is not new. The most prominent example is the cognitive dissonance theory of Festinger [16], which posits that values of chosen options are reinforced to reduce cognitive dissonance between the chosen and the unchosen option. Although early evidence for the theory by Brehm [17] has been challenged on methodological grounds [18,19], more recent studies have provided new support [20–27]. In a very recent study, Luettgau and colleagues [28] have shown that such choice-induced preference changes can also be observed for classically conditioned stimuli.

In the present work, we designed an instrumental conditioning task in which observers learned about the monetary values of a set of conditioned stimuli (CS). Crucially, after an initial training phase with monetary feedback, subjects entered a second phase in which action-contingent feedback was omitted. Subjects were told that they would eventually receive the rewards for their actions at the end of a block, but they did not get trial-by-trial feedback on their choices. We reasoned that, in the absence of external feedback, value representations would still be shaped by a subject's confidence in their choices.

While our main analytic approach was model-based (see below), we also tested three direct behavioral hypotheses. Specifically, we reasoned that if the degree of confidence in a value-based choice reinforces the value of this very choice, the result is a self-reinforcing cycle in which subjective values for more preferred choices are further strengthened and less preferred choices are further devalued. Over time, the absence of external feedback in instrumental conditioning should thus lead to an augmentation of preferences for available choice options. We therefore hypothesized that the absence of feedback would lead to 1) an augmentation of initial preferences ("the rich get richer and the poor get poorer"), and as a result to 2) an increase of choice consistency and 3) an increase of choice confidence (as preferences become more defined).

To better understand the dynamics of value changes in the absence of feedback – and a potential role of confidence therein – we devised a family of computational models in which confidence guides learning when no external feedback is available. In terms of a confidence-based learning signal, we adopted the notion of confidence prediction errors: the difference between expected confidence and actual confidence [12]. We have previously shown that confidence prediction errors constitute a sensible computational learning signal in the context of perceptual learning and that a ventral striatal correlate of this signal was predictive of perceptual learning success [12,29].

## Results

### Behavioral results

The experimental paradigm was structured in the logic of a standard value-based decision making task in which participants had to learn about the values of initially neutral conditioned stimuli (CS). The experiment consisted of 11 blocks in each of which participants had to learn about the value of 5 new CS with different objective values. Trial-wise feedback was provided in the first and third phase (phases 1 and 3) of a block, but critically, was omitted for a varying number of trials in between (phase 2) (Fig 1A). In each individual trial, participants had to make a choice between two CS and subsequently indicated their choice confidence on a scale from 0 to 10 (Fig 1B).

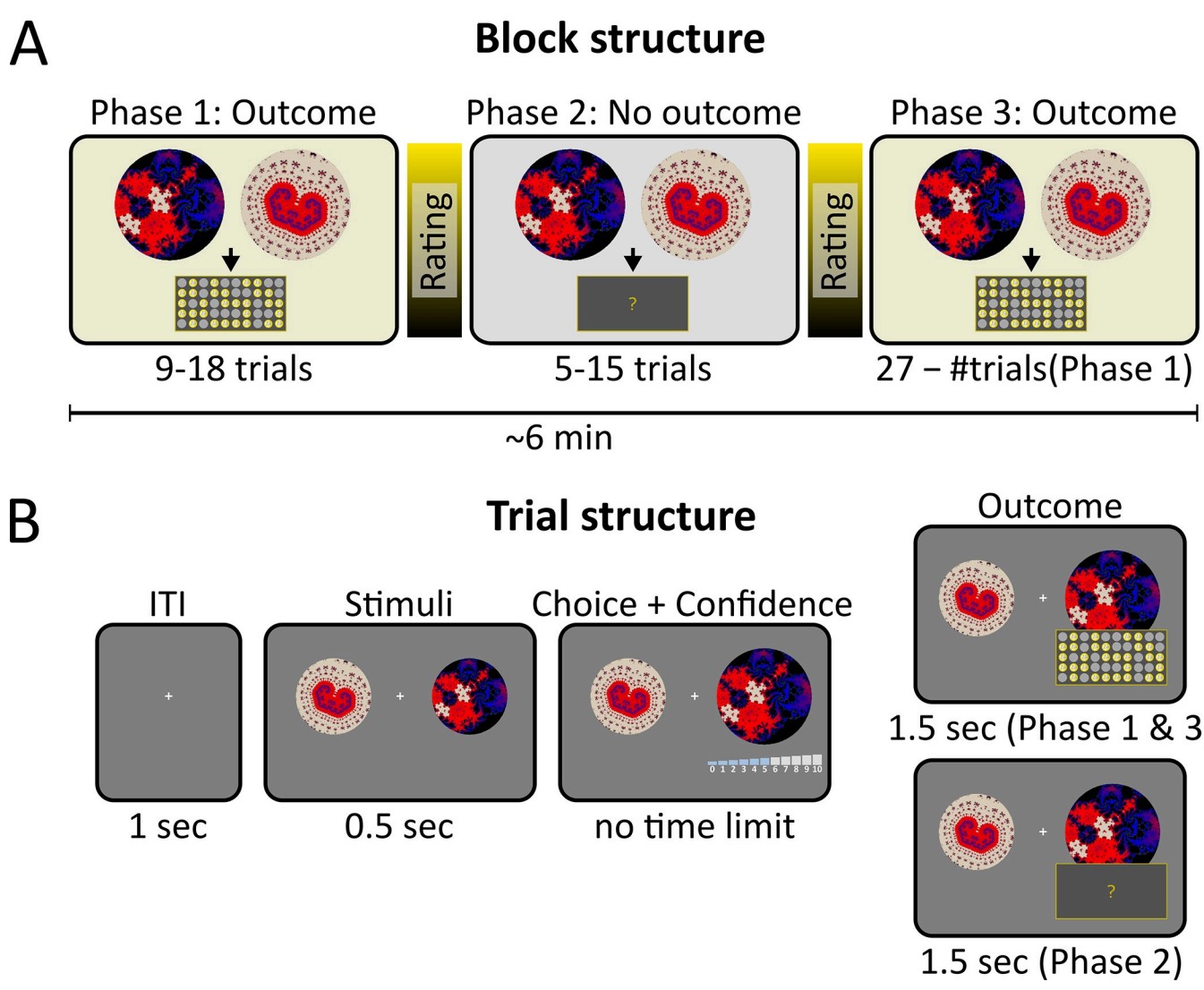

**Fig 1. Experimental design. (A)** Block structure. In each block, participants had to learn about the values of five CS based on the feedback in phases 1 and 3. The critical phase was a period of 5–15 trials in-between phases 1 and 3 in which participants did not receive feedback (phase 2). Before and after phase 2, participants rated the values of each CS on a continuous scale. **(B)** Trial structure. In each trial, participants chose between two CS and indicated their confidence on a scale from 0 to 10. In phases 1 and 3, the reward outcome for the chosen option was presented in the form of a scratch card with 50 fields, each of which could contain a 1 EUR coin or a blank. In phase 2, the scratch card was not revealed; however, participants were instructed that they would receive the hidden reward on the scratch card at the end of the experiment.

We first ensured that participants successfully learned the task. For all analyses involving behavioral learning effects, we used either generalized linear (GLMM; for the correctness of choices) or linear (LMM; for confidence) mixed effects models. We found that participants improved their choice performance (proportion correct) by learning from trial-wise feedback, as indicated by a main effect of trial number across the feedback phases 1 and 3 (GLMM: $z = 11.72$, $p < 0.001$; Fig 2A and Table A in S1 Appendix). In addition, this was reflected in a concurrent increase of subjective confidence across trials (LMM: $z = 68.20$, $p < 0.001$; Fig 2B and Table B in S1 Appendix). Overall, participants' performance increased from $0.63 \pm 0.01$ (s.e. m.) in phase 1 to $0.77 \pm 0.01$ (s.e.m.) in phase 3 (paired t-test: $t_{63} = 13.32$, $p < 0.001$) and their confidence increased from $3.27 \pm 0.25$ (s.e.m.) in phase 1 to $6.06 \pm 0.27$ (s.e.m.) in phase 3 ($t_{63} = 17.26$, $p < 0.001$).

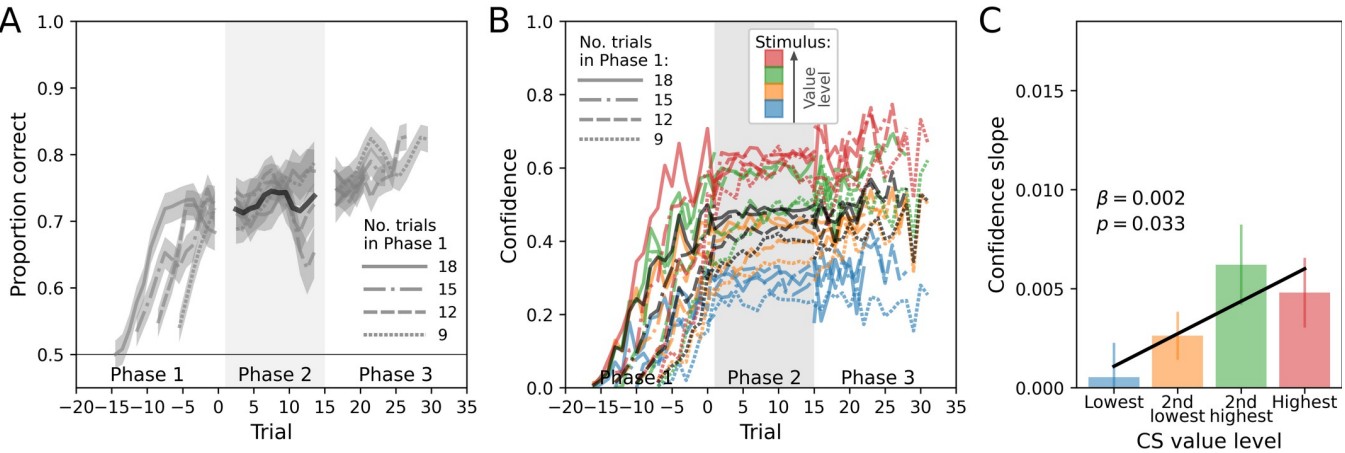

**Fig 2. Performance and confidence.** Block-averaged time courses are separated according to the duration of phase 1 (9–18 trials) and aligned to the beginning of phase 2. Shaded areas indicate standard error of the mean. **(A)** Value-based learning. The accuracy of choices gradually increased across the phases with feedback (phases 1 and 3), indicating that participants successfully learned the task. **(B)** Confidence. Reported confidence (normalized to [0; 1]) likewise increases across the course of a block. Black lines indicate averages across CS value levels. **(C)** Confidence increases in phase 2 in dependence of the CS value level. The parameter estimate $\beta$ and the p-value are based on a linear model with value level as IV and average confidence slope in phase 2 as DV.

The primary focus of our investigation was on the behavioral dynamics in phase 2, in which no feedback was provided. Specifically, we were interested whether behavioral changes across phase 2 in terms of choice consistency (see below), confidence ratings and subjective value ratings showed signatures of self-reinforced learning.

Across trials in phase 2, performance did not change significantly, as shown by a non-significant main effect of trial number (GLMM: $z = -0.35$, $p = 0.726$; Fig 2A and Table C in S1 Appendix). By contrast, confidence increased across phase 2 (LMM: $z = 3.12$, $p = 0.002$; Fig 2B and Table D in S1 Appendix) despite the absence of any new information. The confidence increase in phase 2 was still measurable in phase 3: confidence in phase 3 was higher in blocks including phase 2 ($0.58 \pm 0.03$ [s.e.m.]) compared to control blocks in which phase 2 was omitted ($0.55 \pm 0.03$ [s.e.m.]; $t_{63} = 1.9$, $p = 0.032$).

The increase in confidence in phase 2 was dependent on the overall value level of the chosen CS. A linear model with phase 2 confidence slope as DV and value level as IV indicated a significant positive effect of value level ($\beta = 0.002$, $p = 0.033$). Thus, confidence slopes were on average higher for more valuable CS (Fig 2C).

A second signature of self-reinforced learning is an increase of choice consistency, such that participants become more consistent in their choices when repeatedly being faced with the same pair of CS. Indeed, we found that choice consistency tended to increase in the course of phase 2, indicated by a positive effect of CS pair repetition number (GLMM: $z = 1.85$, $p = 0.064$; Table E in S1 Appendix), where the repetition number $n$ refers to the $n$th repetition of a CS pair in phase 2. Fig 3A visualizes the increase in choice consistency by showing the average choice consistency of participants between the first and second occurrence of a choice pair (blue), as well as between the second and the third occurrence (orange). In particular, the proportion of participants showing perfect choice consistency increased from 19% at the second occurrence to 64% at the third occurrence.

Finally, we tested whether subjective value ratings before and after phase 2 would likewise show a self-reinforcing effect, such that CS with higher objective value would gain subjective value relative to CS with lower objective value. We performed a mixed linear regression analysis with rating change (post- minus pre-phase-2) as a dependent variable and objective

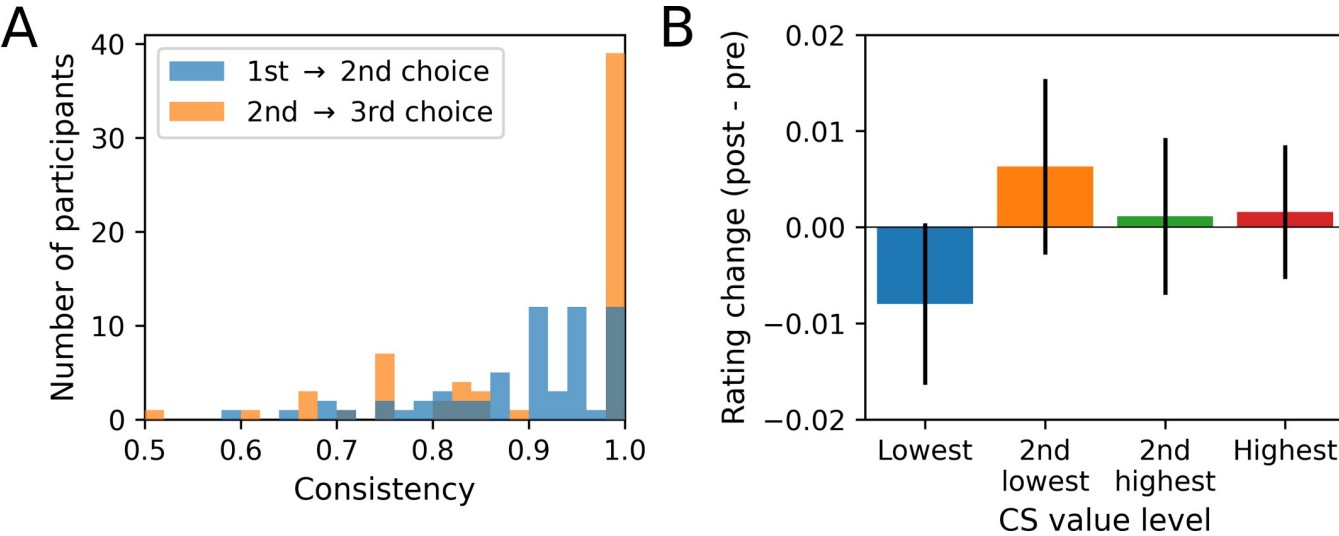

**Fig 3. Changes in choice consistency and subjective value ratings in phase 2. (A)** Choice consistency between first and second (in blue), as well as between second and third choice (in orange) for identical CS pairs in phase 2. **(B)** Subjective value ratings. Depicted are the changes of the subjective value ratings (post-phase-2 minus pre-phase-2), separately for each of the four CS value levels within a block.

stimulus value as our main independent variable of interest. While the effect was in the expected direction, the effect was far from being significant (LMM: $z = 0.64$, $p = 0.522$; Table F in S1 Appendix).

Fig 3B visualizes the rating change as a function of CS value, aggregated by the relative CS value order for simplicity reasons (note that while there were 5 CS per block, they were assigned to 4 distinct value levels). Although the absence of an interaction is apparent, the lowest-ranking CS (here displayed in blue) showed an overall rating decrease, while the higher-ranking CS showed numeric increases. As we will elaborate in the discussion, ceiling effects or regression to the mean effects may have masked a potential interaction. Yet, even in this case, the effect is likely a weak one. In an exploratory analysis, we found that the value dependency of rating changes showed a significant positive interaction with the length of phase 2 (LMM interaction effect: $z = 2.72$, $p = 0.006$; Table G in S1 Appendix and S1 Fig). This suggests that longer phases without feedback lead to a stronger effect of value on rating changes.

## Computational models of value-based learning in the absence of feedback

In line with the neurocomputational similarities between reward- and confidence-based learning [11,12,30], we assume two basic feedback modes. In reward mode, observers maintain a running estimate of *expected values* $\bar{v}_i$ of each stimulus $i$ that is updated by means of a conventional Rescorla-Wagner learning rule. Learning is based on reward prediction errors, i.e. the difference between the reward $r$ that was obtained in a given trial and the expected value $\bar{v}_i$ of the chosen stimulus $i$:

$$\bar{v}_i \rightarrow \bar{v}_i + \alpha_r \, \Delta v \tag{1}$$

$$\Delta v = r - \bar{v}_i \tag{2}$$

The speed of learning is controlled by a *reward learning rate* $\alpha_r$.

Analogously, we assume that observers maintain a running average of the confidence $\bar{c}_{(i)}$ they experienced in past choices of stimuli $i$.

$$\bar{c}_{(i)} \rightarrow \bar{c}_{(i)} + \alpha_c \, \Delta c \tag{3}$$

$$\Delta c = c - \bar{c}_{(i)} \tag{4}$$

Thus, expected confidence $\bar{c}_{(i)}$ is likewise learned and updated by a prediction error signal – in this case the difference between current confidence $c$ and the preceding estimate of expected confidence (confidence prediction error). Crucially, current confidence is a behavioral measure obtained through subjective reports in a given trial. The update speed is controlled by a distinct *confidence learning rate* $\alpha_c$. Note that we put the index ($i$) in brackets to anticipate that we will distinguish between models that update expected confidence in either a stimulus-specific (model *ConfSpec*) or stimulus-unspecific (model *ConfUnspec*) manner. Stimulus-specific models maintain a running estimate of expected confidence for each stimulus separately, whereas stimulus-unspecific models maintain a single stimulus-independent estimate of expected confidence.

Our key hypothesis is that, in the absence of external feedback, value estimates are affected by confidence prediction errors. For instance, when making a choice in which we are very confident, and which thus will typically elicit a positive confidence prediction error, the value of the chosen option is increased. This mechanism is controlled by the *confidence transfer parameter* $\gamma$:

$$\bar{v}_i \rightarrow \bar{v}_i + \gamma \, \Delta c \tag{5}$$

Thus, the value of the chosen option (as predicted by the model) is updated in proportion to confidence prediction errors. Note that while expected confidence is tracked throughout the experiment, we assume that confidence-based value updates only apply when no external feedback is available.

The proposed model differs from the original perceptual learning model [12] in terms of how a diffuse confidence prediction error signal takes effect: in the perceptual learning model, confidence prediction errors shaped the weights of a simple sensory processing network, requiring a Hebbian learning component in Eq (5) to ensure differential effects on signal and noise weights (known as a three-factor learning rule; [31]). Since the present value-based decision task does not involve processing of a perceptually ambiguous stimulus, the model architecture is simpler and requires only the standard one-factor prediction error learning rule of Eq (5).

Overall, the mechanism of self-reinforcement described by Eqs 3–5 augments initial preferences (which might have emerged in a phase with feedback) such that initially more preferred options are further positively reinforced and less preferred options are less reinforced or even negatively reinforced. As a consequence, the value landscape becomes more defined and ensuing choices between choice options are made with higher confidence.

As a first control, we test a model in which the mere act of a choice – without a modulation by confidence prediction errors – leads to a reinforcement of the associated stimulus:

$$\bar{v}_i \rightarrow \bar{v}_i + \lambda \tag{6}$$

This *Choice* model is reminiscent of the idea of choice-induced preferences changes [16], which posits that values of chosen options are reinforced to reduce cognitive dissonance between the chosen and the unchosen option.

**Table 1. Models.**

| Name | Dynamics in the absence of external feedback |
|---|---|
| *Static* | Values are unchanged / static |
| *Deval* | Values of chosen options are subject to devaluation |
| *Choice* | Values of chosen options are reinforced irrespective of confidence |
| *ConfSpec* | Values of chosen options are updated in proportion to stimulus-specific confidence prediction errors |
| *ConfUnspec* | Values of chosen options are updated in proportion to stimulus-unspecific confidence prediction errors |
| *Perseveration* | Choice perseveration bias, but values remain unchanged |

Moreover, we consider the possibility that, in the absence of external feedback, stimuli are subject to devaluation. Although subjects are aware that they will receive the rewards associated with all choices at the end of the experiment, the omission of a choice-contingent reward display might nevertheless cause a devaluation of choice options. This third mechanism is referred to as the *Deval* model and is implemented in a way that subjects perceive the absence of trial-by-trial reward feedback as if they received an effective reward of zero. The reward prediction error thus becomes $0 - \bar{v}_i$:

$$\bar{v}_i \rightarrow \bar{v}_i + \alpha_d \left(0 - \bar{v}_i\right) = \left(1 - \alpha_d\right) \bar{v}_i \tag{7}$$

The speed of devaluation is controlled by a separate devaluation learning rate $\alpha_d$. As before, the update rule only affects the chosen stimulus $i$.

Finally, we tested a model in which choices likewise become more consistent in the absence of external feedback, but in which the actual values are unchanged. This is accomplished by means of a choice perseveration bias parameter $\eta$ [32,33], which captures tendencies to perseverate (positive values) or alternate (negative values). The parameter $\eta$ of this *Perseveration* model affects choice probabilities and is described in the Methods section on 'Model parameters and model fitting'.

In sum, we therefore consider models in which values are either unaffected in the absence of feedback, affected by devaluation, affected by the mere act of a choice or affected by confidence prediction errors (stimulus-specific or -unspecific). Table 1 provides an overview of the models under consideration and Table 2 provides information about the parameters of each model.

**Table 2. Free model parameters.**

| Symbol | Lower bound | Upper bound | Fitted values (mean ± SEM) | | | | | |
|---|---|---|---|---|---|---|---|---|
| | | | *Static* | *Deval* | *Choice* | *ConfSpec* | *ConfUnspec* | *Perseveration* |
| $\alpha_r$ | 0 | 1 | .26 ± .02 | .26 ± .02 | .26 ± .02 | .23 ± .02 | .21 ± .02 | .28 ± .02 |
| $\beta$ | 0 | 2 | .23 ± .03 | .23 ± .03 | .22 ± .03 | .24 ± .03 | .27 ± .03 | .22 ± .03 |
| $\alpha_c$ | 0 | 1 | | | | .14 ± .03 | .14 ± .03 | |
| $\alpha_d$ | 0 | 1 | | .0001 ± .0001 | | | | |
| $\gamma$ | 0 | inf | | | | 6.68 ± .98 | 8.31 ± 1.09 | |
| $\lambda$ | 0 | inf | | | .97 ± .13 | | | |
| $\eta$ | −5 | 5 | | | | | | .73 ± .05 |

## Quantitative model comparison: unspecific confidence prediction errors guide value-based learning in the absence of external reward feedback

While the behavioral analyses provided partial evidence for self-reinforcing effects in the absence of external feedback, they are agnostic about the underlying mechanism. To differentiate between different possible mechanisms, and in particular the role of confidence therein, we statistically compared the models introduced before. Three main research questions were associated with this comparison. First, we aimed to clarify whether a confidence-based learning signal interacts with subjective values and thereby partially explains the dynamics of choices in the absence of external feedback. Second, in the context of confidence-based learning models we were specifically interested in whether the computation of confidence prediction errors relies on a running estimate of expected confidence that is computed in a stimulus-specific (*ConfSpec* model) or stimulus-unspecific (*ConfUnspec* model) manner. And third, we tested whether two simpler models may account for the behavior in phase 2: the *Choice* model, in which subjective values are influenced by the mere act of a choice without a modulation by confidence; and the *Deval* model, in which stimuli are subject to devaluation in the absence of feedback.

We computed the model evidence by means of the Akaike information criterion (AIC; [34]) in order to account for the varying complexity of models. As shown in Fig 4, we found that the *ConfUnspec* model best accounted for the choice dynamics in phase 2. The model evidence of the *ConfUnspec* model was significantly better compared to the evidence for the second-best model, the *ConfSpec* model (paired t-test: $t_{63} = 4.14$, $p < 0.001$), and compared to the *Static* model ($t_{63} = 7.55$, $p < 0.001$). A complementary analysis with the Bayesian information criterion (BIC) confirmed the *ConfUnspec* model as the winning model (S2 Fig).

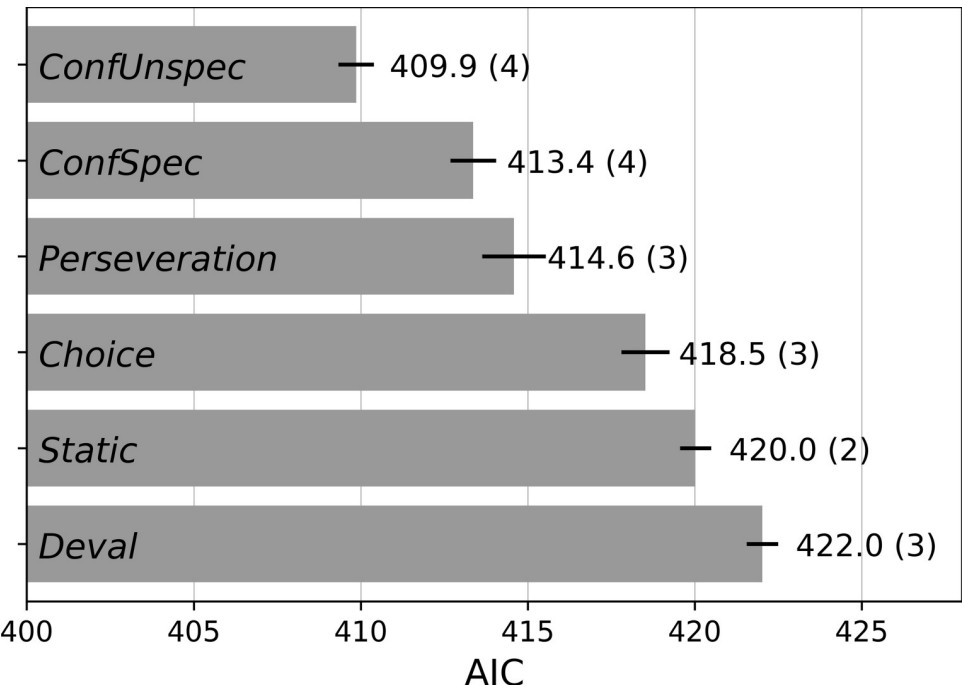

**Fig 4. Model comparison.** Models were compared by means of the Akaike information criterion (AIC). Each value represents the average AIC of a model across participants (± SEM). The number in parentheses indicates the number of model parameters.

Overall, this comparison thus supports our hypothesis that choice dynamics in value-based decision making are partially driven by confidence-prediction-error-based learning signals. Confidence prediction errors are likely computed in reference to a stimulus-unspecific baseline, i.e. only a single estimate of expected confidence is maintained. By contrast, a model in which the mere act of a choice affects subjective values regardless of confidence performed better than an entirely static model, but was clearly inferior to the confidence models. This suggests that choice confidence may be a key variable to consider when examining the effects of choice-preference changes also in contexts other than the present value-based decision making paradigm.

Finally, it is worth pointing out that the evidence against a simple devaluation model was striking. Not only did this model perform worse than the *Static* model, an inspection of devaluation learning rates $\alpha_d$ also revealed that for 96.9% of the participants the best fit for $\alpha_d$ was exactly zero.

## Temporal dynamics of the winning model: latent variables and posterior predictive fits

To get a better picture of the inner workings of the *ConfUnspec* model, we inspected the time courses of latent model variables as well as posterior predictive fits for performance and confidence. The time course of the model's expected value shows how value estimates become more distinct over time and become arranged in the order of objective CS values (Fig 5A). This pattern reflects the fact that, on average, participants successfully learned the task. Notably, due to self-reinforcement the values continue to spread even in the absence of external feedback (phase 2), reflected in a concurrent increase of predicted model performance (Fig 5B).

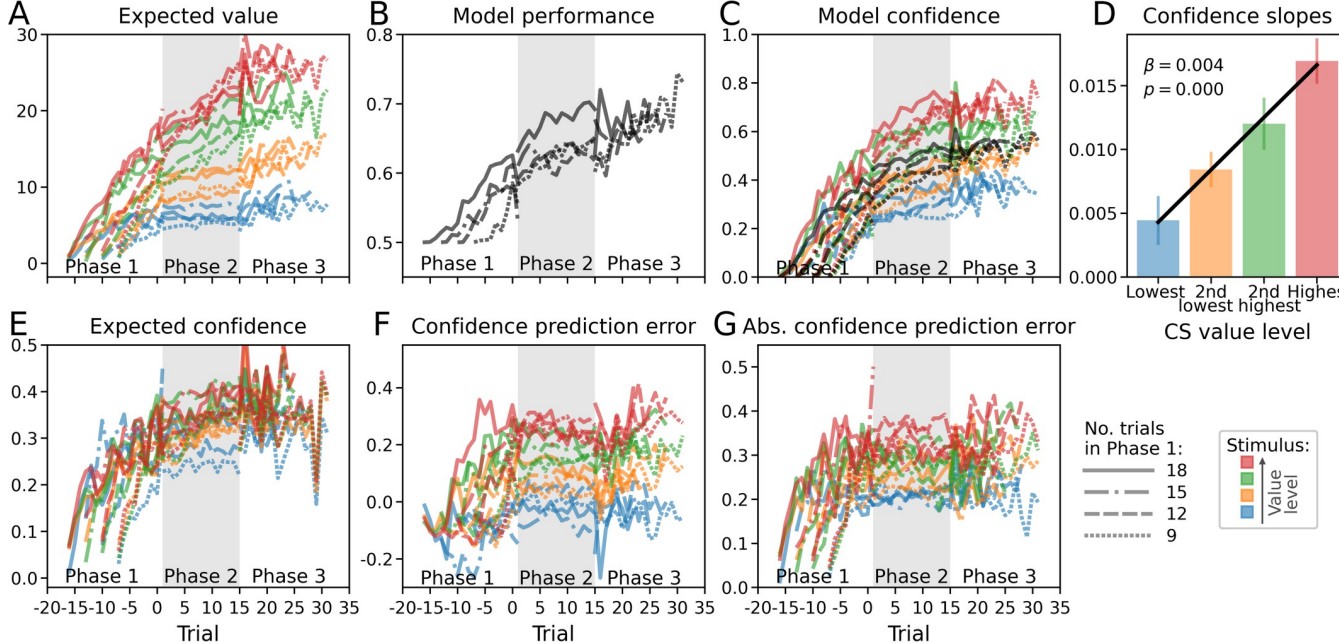

**Fig 5. Latent variables and posterior predictive fits of model *ConfUnspec*.** All time courses represent averages across blocks and subjects, split according to the duration of phase 1 (line styles) and the four CS value levels within a block (colors). **(A)** Expected values indicate current beliefs about the value of each stimulus. **(B)** Posterior predictive fit for model performance: expected proportion correct responses based on choice probabilities. **(C)** Posterior predictive fit for model confidence. Model confidence is computed based on the choice probability for the chosen CS (normalized to the range 0–1). Black lines indicate averages across value levels. **(D)** Confidence slopes of (C) in phase 2 in dependence of the CS value level. **(E)** Expected confidence corresponds to an integration of past confidence experiences using a Rescorla-Wagner-type learning rule. **(F)** Confidence prediction errors indicate the deviation of a momentary confidence experience from expected confidence. **(G)** Absolute confidence prediction error.

To assess the posterior predictive fit for confidence, we computed model confidence as $2 \cdot (p_{\text{choice}} - 0.5)$ to ensure the same range 0–1 as for normalized behavioral confidence ratings. As to be expected, the model's confidence predictions likewise show an increase across phase 2 (Fig 5C). Moreover, confirming the behavioral results, the confidence increase is dependent on the overall CS value level (Fig 5D). This result is independent evidence that the metacognitive dynamics at the behavioral level are subject to a self-reinforcement mechanism.

Regarding the latent confidence variables we found that expected confidence likewise increases over time, in line with the increase of confidence (Fig 5E). For expected confidence, the differentiation with respect to the objective CS values is also evident, although less pronounced than in the case of expected value. It is noteworthy that confidence prediction errors, on average, are positive in phase 2 for all but the lowest-value CS (Fig 5F). One reason is that the learning rates for expected confidence ($\alpha_c$) are relatively small for quite a few participants (cf. Fig 6D), such that expected confidence reflects the increase of confidence only with a delay. For those participants the expected confidence only slowly increases from its initial value of zero and thus the learning signal (confidence minus expected confidence) is well

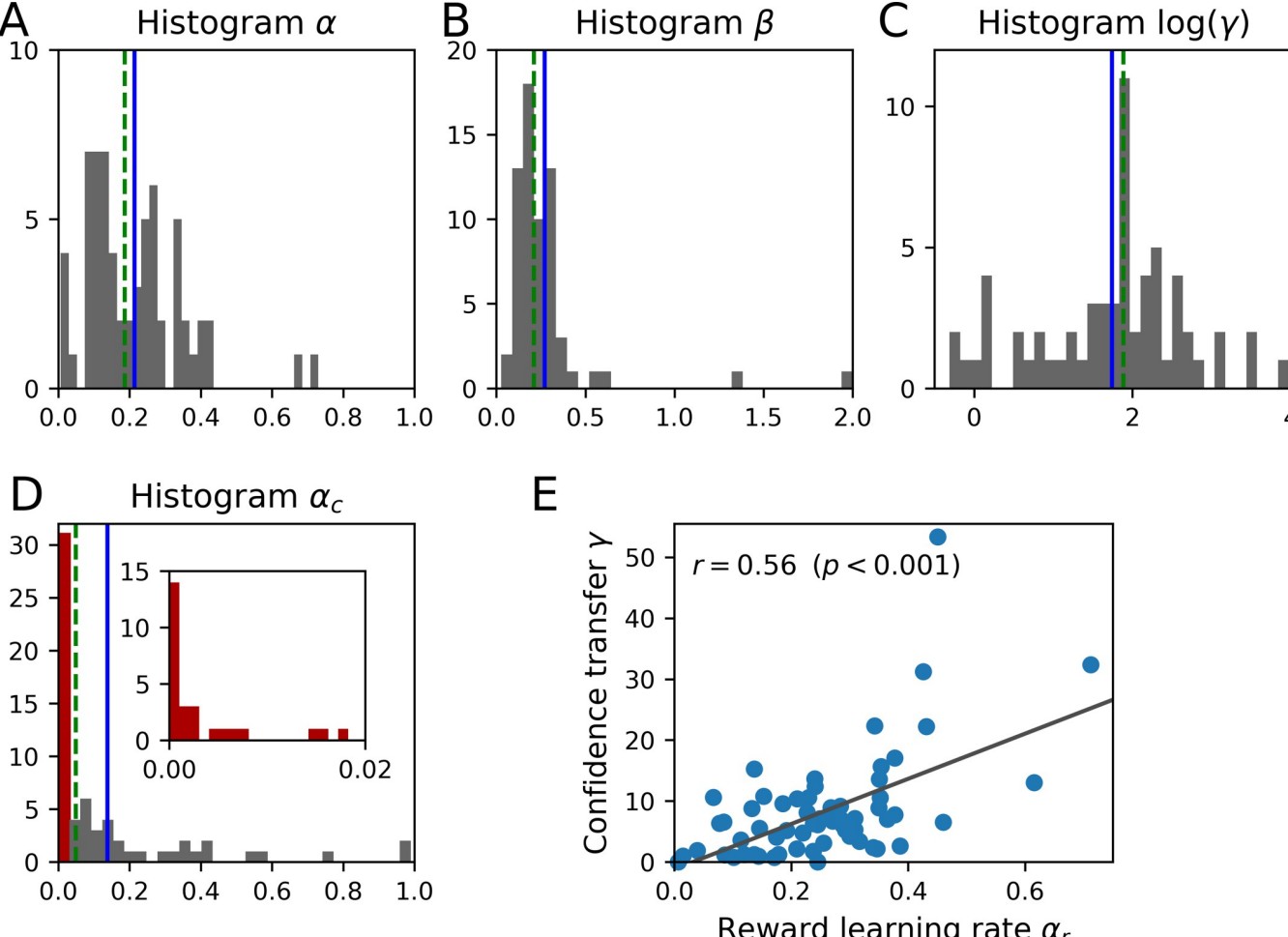

**Fig 6. Model parameters for the winning model *ConfUnspec*.** Blue solid lines indicate parameter means, green dashed lines parameter medians. **(A)** Histogram of the reward learning rate $\alpha_r$. **(B)** Histogram of the inverse decision noise parameter $\beta$. **(C)** Histogram of the confidence transfer parameter $\gamma$. **(D)** Histogram of confidence learning rate $\alpha_c$. **(E)** Scatter plot between reward learning rate $\alpha_r$ and confidence transfer $\gamma$. The black line indicates a linear fit to the data; the correlation coefficient is based on a Pearson correlation.

approximated by confidence itself for a certain setting period. A likely second reason is that the confidence value transfer ($\gamma$) of positive CPEs itself triggers a self-reinforcing cycle: positive CPEs increase the value of the chosen CS and thus the confidence in future choices of this CS, which in turn increases the probability of positive CPEs.

## Qualitative model comparison between self-reinforcing models

To qualitatively compare the *ConfUnspec* model to the two other self-reinforcing models and to the *Perseveration* model, we first assessed the latent variables and posterior predictive fits for performance and confidence of these models as well. The second-best model, *ConfSpec*, has a confidence-based mechanism as well. As shown in S3A, S3B and S3E Fig, the latent variables of *ConfSpec* – expected value, expected confidence, confidence prediction error – show largely the same patterns as for *ConfUnspec*. The main difference is that the median confidence learning rates $\alpha_c$ are, on average, lower for *ConfSpec*, which leads to a slower build-up of expected confidence; this in turn results in larger average and absolute average prediction errors compared to the winning *ConfUnspec* model (cf. Fig 5F and 5G) and thus more variable learning signals. The fact that the time course of expected confidence does not timely track actual confidence may be an indication that the *ConfSpec* is a less adequate fit to the data than *ConfUnspec*.

In terms of behavioral predictions for subjective values, performance and confidence, the winning model *ConfUnspec* (Fig 5) is largely indistinguishable from the models *ConfSpec* (S3E–S3G Fig) and *Choice* (S3I–S3K Fig). In particular, all three models predict a spreading of subjective values in phase 2, and as a consequence, an increase of confidence. As observed for the behavioral data and the winning *ConfUnspec* model, *ConfSpec* and *Choice* likewise predict that the increase of confidence in phases without feedback is value-dependent (S3H and S3L Fig). In contrast, while the *Perseveration* model does not predict changes in confidence in phase 2, it is the only model that predicts a flat type 1 performance curve. The reason is that the *Perseveration* model leaves expected values unchanged and merely causes choices to become more consistent (the perseveration bias $\eta$ is positive for all participants; cf. Table 2). On the other hand, the fact that the self-reinforcing models predict an increase in performance (Figs 5B and S3F, S3J) has to be acknowledged as an incorrect prediction of the self-reinforcing models. Overall, the posterior predictive fits of our models draw a somewhat opaque picture as to why specifically the *ConfUnspec* model outperforms the two other self-reinforcing models.

To clarify whether the behavioral predictions of the models naturally arise from the models or are caused by overfitting, we performed a second analysis in which we more generally assessed model generative performance for a variety of parameter settings (S4 Fig). As for the behavioral analyses, we assessed changes of confidence and performance in phase 2 (referred to as *Confidence effect* and *Performance effect*), the interaction of confidence slopes and value (*Confidence x value effect*) and changes of choice consistency in phase 2 (*Consistency effect*).

We found that two of the behaviorally observed effects are robustly produced by all self-reinforcing models (i.e. *ConfUnspec*, *ConfSpec*, *Choice*) across parameter regimes: a positive *Confidence effect*, i.e. an increase of confidence across phase 2, and a positive *Consistency effect*, corresponding to an increase of choice consistency. By contrast, the *Confidence x value effect*, i.e. higher confidence slopes for more valuable CS, emerges as a general effect only for the confidence-based models *ConfUnspec* and *ConfSpec*. In addition, the *ConfSpec* model exhibits this effect only for small values of $\alpha_c$, whereas in the *ConfUnspec* model the effect arises robustly across different settings of $\alpha_c$. Coincidentally, as noted above, we found that the fitted parameters for $\alpha_c$ were lower (and more frequently close to zero) in the *ConfSpec* model compared to the *ConfUnspec* model. While not allowing for strong conclusions, this observation indicates that the empirically observed *Confidence x value effect* arises most naturally for the *ConfUnspec*

model and can only be achieved by special parameter combinations for the *ConfSpec* model. This may be a potential explanation for the superior model fit of *ConfUnspec*.

Compared to the *Confidence effect*, the positive *Performance effect* (which was absent in the empirical data) only emerges under specific parameter combinations for the *ConfSpec* and *ConfUnspec* model. Specifically, the *Performance effect* arises precisely for parameter combinations that also yield a *Confidence x value effect*. This makes sense: when observers become more confident in more valuable CS this should go hand in hand with an increase in performance. Importantly, the *Performance effect* is minuscule with slopes in the order of at most 0.001 (corresponding to a performance increase of 1% across 10 trials). This could explain why no performance effect was observed in the behavioral data, even if behavior was shaped by a confidence-based reinforcement mechanism. Finally, note that the *Perseveration* model does not show any of the discussed effects to a meaningful degree within the assessed parameter ranges.

### Relationship between reward-based and confidence-based learning

We reasoned that, if learning with and without external feedback is based on a similar mechanism, interindividual differences in reward-based learning may be predictive of interindividual differences in confidence-based learning. While reward-based learning is characterized by the reward learning rate $\alpha_r$, the impact of confidence on subjective values is captured by the confidence transfer parameter $\gamma$. Fig 6 shows the distributions of parameters for $\alpha_r$ and $\gamma$, as well as for the two remaining parameters of the winning model, namely decision noise $\beta$ and confidence learning rate $\alpha_c$.

We indeed found a strong correlation between the reward learning rate $\alpha_r$ and the confidence transfer parameter $\gamma$ in our winning model ($r = 0.52$, $p < .001$). As a control analysis, and to ensure that both estimates are independent of one another, we correlated the reward learning rate of the *Static* model to the confidence transfer parameter. Here, again, the effect holds, with $r = 0.56$, $p < .001$ (Fig 6E). Thus, observers who show more volatile reward-based updating of their value-based beliefs also show higher volatility for learning based on confidence prediction errors, when feedback is no longer provided. Of note, the reward learning rate $\alpha_r$ was not correlated to the speed with which observers updated their estimates of expected confidence, characterized by the confidence learning rate $\alpha_c$ ($r = -0.09$, $p = 0.498$; control analysis with *Static* model: $r = -0.03$, $p = 0.842$).

## Discussion

We investigated the role of confidence-based learning signals in value-based learning and decision-making when external feedback is not available. Consistent with our hypothesis, we found behavioral evidence for signatures of confidence-based self-reinforcement: an increase of subjective confidence, increased choice consistency and a tendency towards self-reinforcement of subjective values. A model-based analysis showed that a model which considered confidence-based learning signals in phases without external feedback outperformed a static model, as well as a model that predicted devaluation over time.

Overall, our findings thus corroborate the notion that confidence reflects an internal reinforcement learning signal, connatural to reinforcement signals induced through external reward or cognitive feedback. The general mechanistic idea therein is that the brain triggers global reward signals when actions or percepts yield higher confidence than expected, thereby reinforcing underlying neural circuits that gave rise to these actions or percepts. For instance, when practising an instrument, internal reinforcement signals may be triggered when the musician is more confident in a particular performance than expected on the basis of previous

attempts. In the context of perceptual learning, such signals may reinforce specific sensory processing pathways that happen to generate percepts associated with above-average confidence.

While the advantage of confidence learning signals is intuitive in these examples, the adaptive advantage of confidence-based learning is less clear in the context of value-based decision making: why should subjective values change at all in the absence of new information?

One possibility is that confidence effects observed in value-based decision making are an accidental side effect – an epiphenomenon – of a mechanism that otherwise proves advantageous in the majority of learning scenarios. In this case, one may seek in vain for the benefits of confidence-based learning in the specific case of value-based decision making. However, another possibility is that self-reinforcement of subjective values may be a pragmatic strategy in the face of a possible memory leakage when feedback is omitted. A classic example for such leakage is retrieval-induced forgetting, i.e. the observation that our memories for items become imprecise merely due to the mnemonic retrieval of these items [35–37]. In line with this notion, a recent study has shown that the mere act of a choice between CS induces changes to hippocampal representations of stimulus-outcome associations [28]. Thus, without external feedback subjective values of stimuli may become noisy and thus less reliable, at least when observers continue to interact with these stimuli.

In this latter view, confidence-based self-reinforcement of subjective values could be a counter strategy for memory loss, trading a more black-and-white estimate of the value landscape (a result of self-reinforcement) with the alternative of an overall flattened landscape in which choices become more indifferent (a result of unsystematic noise). In other words, while it may seem irrational when choice options are transformed into a simplified categorical scheme of either good or bad options, such a scheme may actually be more robust towards mnemonic deterioration. Indeed, in the absence of a memory loss mechanism, the generative performance analysis of the winning model *ConfUnspec*, and under more specific settings also the *ConfSpec* model, indicates that self-reinforcement can even lead to a (potentially compensatory) increase in performance as values become more defined (cf. S4 Fig). Such a scenario could explain the behavioral effects of stable type 1 performance with a parallel increase of confidence and choice consistency.

Contrary to our expectation, we did not find a significant value by value change interaction for the subjective value ratings before and after the phase without feedback (although the general direction of results are consistent with our hypothesis). We consider two possible effects that may have counteracted a value-dependent increase of subjective values in phase 2. First, participants were instructed to use the continuous rating scale in an intuitive manner. Naturally, subjects therefore tended to select the lowest and highest ratings for the CS they regarded least and most valuable, respectively. However, in many cases, this intuitive usage of the rating scale effectively left little room for even lower or higher post-phase-2 ratings. Thus, the hard constraints imposed by the scale may represent a systematic bias in the opposite direction of our hypothesis.

Second, the possibility of noisy memory leakage over the course of phase 2 is expected to lead to a regression to the mean for all CS. Although our proposed mechanism is thought to mitigate this leakage, the regression-to-the-mean effect is likewise in the opposite direction of our hypothesis and thus reduces the sensitivity to find the interaction. Higher statistical power is necessary to clarify whether the observed null effect is real or a consequence of insufficient statistical power. Alternatively, it is possible that participants simply were not aware of the subtle value changes occurring in phase 2 and hence these changes were not reflected in the subjective ratings.

In the logic of the best-fitting computational model (*ConfUnspec*), subjective values of chosen CS are reinforced if, and only if, choice confidence is higher than expected on the basis of

previous confidence experiences, i.e. in the case of positive confidence prediction errors. By contrast, chosen CS are devalued if confidence prediction errors are negative. It is noteworthy that the *ConfUnspec* model, i.e. a model with an unspecific reference (expected confidence) to which momentary confidence levels are compared, outperformed a model in which expected confidence was CS-specific. We considered this unlikely, a priori, since an unspecific reference deprives the confidence prediction error from its natural convergence property: an unspecific reference maintains the average confidence level across all CS so that prediction errors, in principle, can be persistently positive (for CS judged to be of relatively high value) or negative (for CS judged to be of relatively low value).

However, the similarity of posterior predictive fits between the self-reinforcing models *ConfUnspec*, *ConfSpec* and *Choice* (Figs 5 and S3A–S3L) gives reason to be cautious about a specific mechanistic interpretation of the behavioral effects. While only the posterior predictive fits of the three self-reinforcing models showed the *Confidence x value effect*, there is no 'smoking gun' that discriminates between these models. Our analysis of model generative performance provided only partial resolution in that the *Confidence x value effect* was found not to be a natural property of the *Choice* model and is produced only under specific parameter settings for the *ConfSpec* model. Thus, the *Confidence x value effect* arises most naturally in the *ConfUnspec* model. The fact that this effect is more robust across different parameters for the *ConfUnspec* model means that these parameters have more flexibility which could be a potential explanation for the superior model fit.

The present work thus provides evidence that value-based learning in the absence of external feedback is shaped by some form of self-reinforcement, but the *specific* proposed mechanism of the *ConfUnspec* model is mainly supported by our quantitative model comparison and not by a clear falsification of the other self-reinforcement models. Only the *Choice* model falls off to a certain degree as it does not generally produce a *Confidence x value effect* and thus doesn't support our proposed adaptive mechanism of self-reinforcement, i.e. protecting against memory leakage in phases where subjective preferences are not refreshed by external feedback. Combined with the fact that the *Choice* model did not perform well in the quantitative model comparison, ranking even behind the *Perseveration* model, we suggest that a confidence-based rather than a mere choice-based self-reinforcement mechanism is likely. Disambiguating between the two confidence models might necessitate an experimental paradigm that is tailored to the differences between these models, for instance by introducing conditions that manipulate the degree to which the unspecific prediction error references of the *ConfUnspec* model are problematic for learning (e.g., 'roving' conditions; [37]).

A key parameter in both confidence-based models is the confidence-transfer parameter $\gamma$, which controls the degree to which confidence prediction errors affect subjective values when no external feedback is available. By contrast, in the case of external feedback, the update of subjective values is based on reward prediction errors and governed by the learning rate parameter $\alpha_r$. Intriguingly, we found that both parameters are strongly correlated, such that participants with more volatile reward-based value learning also showed more volatile confidence-based value learning.

This finding fits well with our motivating hypothesis that learning based on external reward feedback and internal confidence-based feedback share similar – perhaps the same – underlying mechanisms. The parameters $\gamma$ and $\alpha_r$ thus may both characterize the tuning of one and the same learning machinery, observed in scenarios with and without external feedback. Together with the observed neurobiological parallel of learning based on internal and external feedback [10–12], the shared algorithmic logic of the respective learning signals [12,38,39], and the shared phenomenology [15], this parametric correspondence adds another piece of evidence to the view that confidence-based learning is based on an internally-triggered reinforcement learning mechanism.

Our results may have an interesting implication for one of the most prominent and contro-versial effects in the decision-making literature – choice-induced preference changes [16,17,19]. Here too, changes in subjective values are induced in the absence of external feed-back, putatively caused by the mere act of the choice itself. Surprisingly, to our knowledge, almost no study has yet examined the role of choice confidence in choice-induced preference changes (for an exception, see [40]). Indeed, taking Festinger's idea of cognitive dissonance as a cause of these preference changes seriously, it would predict a role of confidence that is in opposition to our model.

According to Festinger, subjective values are increased for chosen options (and decreased for unchosen options) as a form of post-hoc rationalization, to reduce the dissonance that would arise otherwise when reflecting on the positive attributes of an unchosen option. The larger the dissonance, the stronger the expected preference changes. Since the dissonance will be stronger for choices that are subjectively perceived as harder, those choices should be asso-ciated with a lower level of choice confidence. Thus, Festinger's theory predicts that higher choice confidence leads to higher preference changes for the chosen option, whereas our pro-posed model predicts the opposite (note however, that our model does not consider changes for the unchosen option). It will be an interesting avenue for future research to systematically investigate the interplay of choice confidence and subjective values changes and thereby clarify which prediction best passes the empirical evidence. Our finding suggests that choice confi-dence is a key variable to consider in this question.

An assumption made in the present study is that self-reinforcement is restricted to instances without external reinforcement or cognitive feedback. However, this assumption was not explicitly tested and, at least from a conceptual point of view, the proposed self-rein-forcement mechanism in our models could be readily implemented as a modulation of (exter-nal) feedback-based model updates, or as a mechanism parallel to those. To test this possibility experimentally, one could introduce an alternative phase 2 that is matched in every respect except the fact that external feedback is provided.

A limitation of behavioral results is that most effects are not very strong, including an absent main effect for the predicted change of subjective values, which was significant only for the longest duration of phase 2. This suggests that self-reinforcement effects in the absence of external feedback are relatively subtle, or, more unfavourably for the present study, a false posi-tive. Either way, it is clear that investigating the choice and confidence dynamics in the absence of external feedback calls for large sample sizes. Moreover, our results suggest that the emer-gence of self-reinforcement effects at the level of conscious report might require no-feedback phases of sufficient length.

In conclusion, our study provides evidence that confidence-based learning signals can explain significant dynamics of value-based decision making in the absence of external feed-back, thereby extending previous findings in the specific domain of perceptual learning to one of the most fundamental forms of human learning: instrumental conditioning. Our results indicate that a previously suggested conceptual and algorithmic parallel between reward-based feedback and cognitive feedback (e.g., "correct'/'incorrect"; [30]) may have to be extended to internal cognitive feedback – confidence – as well.

## Methods

### Ethics statement

Ethical approval for this study was granted by the ethics committee of Charité, Universitätsme-dizin Berlin. Written informed consent was obtained from all participants prior to the experiment.

## Participants

Sixty-six healthy volunteers (age: 29 ± 8.4 [s.d.]; gender: 40 female) were recruited via online advertisement and word of mouth. Participants were 18 or above and had normal or corrected to normal vision. Their participation was remunerated depending on performance (on average 16.25€). Two participants were excluded due to low task performance (<55% correct responses). The sample size calculation was based on a forward simulation. Choices and confidence ratings (based on the choice probability) were sampled from the generative models using the number of blocks and trials of the empirical experiment. We used educated guesses for all parameters ($\alpha_r = 0.1, \alpha_c = 0.1, \alpha_n = 0.1, \beta = 1/3, \gamma = 1$; disclosure: the model *Perseveration* was tested post-hoc). The sample size was determined such that the model evidence (AIC) of all non-static models could be significantly dissociated from the static model with at least 80% probability (using a two-tailed paired t-test).

## Mixed effects modeling

All analyses involving behavioral learning effects were performed with mixed effects models as implemented in the Python package *statsmodels* (for linear models;[41]) and the *lme4* and *lmerTest* packages in R (for logistic models). *Subject* was a random effect and *block* a nested random effect. Fixed effects were the block-level predictors *block_value_level* (18, 23 and 28, i.e. the overall value level in a block), *block_difficulty* (3 or 6, i.e. the average absolute value difference in a block), *block_stimulus_type* (0 or 1, i.e. stimulus types fractals or Chinese symbols), *block_ntrials_phase1* (duration of phase 1) and *block_ntrials_phase2* (duration of phase 2). Trial-level predictors were *trial_number*, *trial_difficulty* (the absolute value difference between the two CS in a trial) and *trial_value_chosen* (i.e. the value of the chosen CS in a trial).

## Experimental task and procedure

The instrumental conditioning task consisted of 11 blocks with an identical structure (Fig 1A). In each block, participants had to learn about the monetary values of five new conditioned stimuli (CS). Each block started with an initial training phase (phase 1) of variable length (9, 12, 15 or 18 trials) in which feedback was provided. The training phase was followed by a critical second phase (5, 10 or 15 trials) without feedback. In two blocks, phase 2 was omitted as a control condition. At the beginning of phase 2, participants were informed that no feedback would be provided after choices, but also, that they would receive the associated rewards at the end of the experiment. A block was completed by a third phase in which feedback was again provided. The duration of phase 3 was such that, together with phase 1 and phase 2, each block comprised exactly 27 trials.

In each trial (Fig 1B), participants were presented with a choice between two CS on the left and right of a fixation cross, respectively. To choose e.g. the left CS, participants moved the mouse cursor to the left. The choice movement activated a 11-point confidence scale that appeared under the chosen CS. The confidence scale consisted of 11 bars of increasing height (maximum height for maximum confidence). Each bar was labeled with the respective rating (0 to 10). In addition, the first and last bar, corresponding to the minimum and maximum confidence rating, were labeled with "Guessing" and "100% sure". Higher confidence could be indicated by moving the mouse further to the left (or right, when the right CS was chosen), which highlighted all bars up to the respective confidence level. To make the choice/confidence experience more plastic, the CS increased in size proportional to the selected confidence. Participants could still switch their choice during the confidence selection by clicking the right mouse button, although this was rarely the case. When participants were satisfied with their

response, they clicked the left mouse button. At this point, the unchosen CS disappeared and the chosen CS remained on the screen for 1000ms.

In phases 1 and 3, participants received monetary rewards for their choices. Rewards were presented in the form of a scratch ticket with 50 initially grey fields. The 50 fields were successively, but quickly, revealed such that each field was either a blank (in which case the field remained grey) or a hit (in which case a 1-Euro coin appeared on the field). We chose this reward presentation style – over a more conventional reward display with explicit numbers – to induce a mere "feeling" for the value of the CS rather than an explicit cognitive representation of rewards. The revealed scratch card remained on the screen for 500ms and then disappeared in an indicated slit below the card. The presentation in phase 2 was similar except that the fields of the scratch card were not revealed. At the end of the experiment, the overall reward was determined by means of 33 draws from an imaginary lottery box, which comprised all 1-Euro coins and blanks (including those from phase 2 which were initially not revealed) collected during the experiment. The average reward was 16.25€ (SEM 1.64€).

To avoid a learning transfer between blocks, different reward schedules were applied (and indeed there was no main effect of block on performance, p > 0.5). First, each block was assigned one of three different overall average reward levels (18, 23 and 28€ per scratch card). Second, the mean value difference between CS in a block was either 3€ or 6€, which affected the average performance (3€: 68.8% correct; 6€: 77.1% correct). And third, in each block two CS were of identical value. Specifically, there were four different possible values per block to which the five CS were randomly assigned. Rewards were drawn from a truncated normal distribution with the given mean for a CS and a standard deviation of 10€. Since together, these conditions constitute more possible combinations than blocks, the conditions were pseudo-randomly distributed across the blocks. Similar to the variable phase durations, the main purpose was to prevent participants from learning about the task or reward structure and thus to enforce 'learning from scratch' in each block.

In half of the blocks, the CS were multicolor fractals, in the other half monocolor Chinese symbols. There was no meaningful performance difference between the stimulus types (fractals: 72.2% correct; Chinese symbols: 73.6% correct). The size of the CS was between 10.7 and 12.8 degrees of visual angle depending on the confidence level. All CS appeared roughly an equal number of times in each phase of a block.

Before and after phase 2, a rating scale appeared in which participants rated the subjective value of each CS in the current block on a continuous scale. The extremes of the scales were labeled with a scratch card of only blanks (lower end) and only 1-Euro coins (upper end). The scale itself was a horizontal bar with a color gradient from black (lower end) to gold (upper end). To select their rating, participants moved a thin sliding vertical bar across the rating scale (using the computer mouse).

The experiment was programmed in Python using PsychoPy [42]. The experiment took place in a moderately lighted laboratory room in front of a computer screen (1920x1080 pixels, 47.7x26.8cm; viewing distance: 60cm). The entire experiment was operated by a computer mouse.

## Model parameters and model fitting

The model was fitted for each subject individually, using all 11 blocks of the experiment. In the beginning of each block of the fitting procedure, the latent variables *expected value* $\bar{v}_i$ and *expected confidence* $\bar{c}_{(i)}$ were initialized to zero, given that new CS appeared in each block. The choice probability in each trial was computed via a softmax action selection rule [43]:

$$p_{right} = \frac{1}{1 + e^{-\beta(\bar{v}_{right} - \bar{v}_{left})}} \tag{8}$$

$$p_{left} = 1 - p_{right} \tag{9}$$

where $p_{right}$ and $p_{left}$ are the choice probabilities for the CS left and right of the center, respectively. The slope $\beta$ of the logistic function, also referred to as the *inverse decision noise parameter*, accounts for the stochasticity of choices. A value $\beta = 0$ implies that agents respond completely at random, whereas higher values of $\beta$ indicate that agents choose more deterministically the CS associated with the highest expected value.

Importantly, the choice probability in Eq 8 was also used to determine the CS to which the confidence-value transfer (Eq 5) was applied during model fitting (CS$_{right}$ if $p_{right} \geq 0.5$ else CS$_{left}$). Updating the CS actually chosen by the participants would have not been valid, as in this case the model would have had access to the same information it aims to predict.

In case of the *Perseveration* model, the choice probability in the absence of external feedback contains a perseveration bias as follows [32,33]:

$$P_{right} = \frac{1}{1 + e^{-\beta[(\bar{v}_{right} - \bar{v}_{left}) + \eta(c_{right}^{prev} - c_{left}^{prev})]}} \tag{10}$$

where $C_{left/right}^{prev} = 1$ if the CS corresponding to the CS on the left/right side was chosen in the previous encounter of the CS pair ($C_{left/right}^{prev} = 0$ otherwise). An individual with a positive/negative parameter $\eta$ would have a bias towards repeating/alternating the previous response.

Parameters were fitted by minimizing the negative log-likelihood (based on Eqs 8 and 9) using the optimize.minimize() function of the Python SciPy package [44] in combination with an initial coarse-grained grid-search to determine initial values for each parameter. We computed two optimization SciPy routines in parallel – the gradient-based L-BGFS-B algorithm [45] and the conjugate-direction-based Powell algorithm [46] – and chose the parameters of whichever method resulted in a smaller negative log-likelihood.

Table 2 provides an overview about imposed bounds for all parameters. Note that while the learning rate parameters $\alpha_r$, $\alpha_c$ and $\alpha_d$ are bound to the range [0; 1], the *confidence transfer parameter* $\gamma$ is not a learning rate and thus has no natural upper bound. Note that the two new parameters of the model proposed here, the confidence parameters $\alpha_c$ and $\gamma$, were largely uncorrelated (winning model: $r = -0.05$, $p = 0.712$), indicating that neither of them was redundant.

## Model and parameter recovery

To ensure that all models are identifiable with sufficient precision, we performed a model recovery analysis. For each model we simulated choice and confidence data for a range of different parameter settings. $k^N$ different parameter configurations were implemented, where N is the number of parameters in each model (see Fig 4) and k is the number of different values implemented for each parameter ($\alpha_r/\alpha_n/\alpha_c$: range 0.1–1, equidistant steps, k = 5; $\beta$: range 0.1–1.6, doubling steps, k = 5; $\gamma$: range 1–100, exponential steps, k = 5; $\lambda$: range 0.5–5, exponential steps, k = 5; $\eta$: range -1.5–1.5, equidistant steps, k = 6). For each model and parameter combination, we simulated 250 datasets (i.e. subjects). Experimental designs for each dataset were randomly sampled from the design generation function that was used in the behavioral experiment.

Model recovery was quantified by the probability that datasets generated with a given model X were best fitted by a model Y, p(fit = Y|gen = X), as well as the reverse probability that datasets best fitted by a given model Y were generated by a model X, p(gen = X|fit = Y). To obtain p(fit = Y|gen = X), for all datasets created with a given generative model X, we computed the frequency with which model Y had the lowest AIC value among the competing

models (each model was fitted to each dataset). Conversely, to obtain p(gen = X|fit = Y), for all datasets that were best-fitted by model Y (i.e. lowest AIC value), we computed the frequency with which datasets were generated by model X. For p(gen = X|fit = Y), we made sure that the base rate was identical across models, i.e. that the number of datasets generated was equal for all models despite differences in the number of model parameters (and thus in the number of combinatorial parameter settings).

S5 Fig shows the results of the model recovery analysis, expanded for different values of $\alpha_r$ and $\beta$, the two parameters that are common to all models. Overall, we find excellent model recovery. More precisely, p(fit|gen) is generally much higher when the generative and fitting model are identical relative to when they are different. Perhaps even more importantly, the reverse probability matrix, p(gen|fit), likewise demonstrates good model identifiability (S6 Fig). That is, datasets that are best fitted by a given model are, in all likelihood, also generated by this model. This latter analysis is an important prerequisite for any conclusion about underlying mechanisms in our empirical data that derive from the superior model fit of the *ConfUnspec* model.

Notably, the model that performs worst in terms of model identifiability is the *Static* model, both in terms of p(fit|gen) and p(gen|fit). In quite a few instances, datasets generated by the *Static* model are confused with one of the other models (except the *Deval* model). This shows that due to random variation and limited trial numbers (which matched the empirical experiment), the choice dynamics are sometimes better described by more complex models; conversely, the *Static* model is sometimes the best-fitting model although the data is generated by more complex models (because the dynamics generated by the complex models are not distinct enough to compensate for the complexity punishment). Importantly, instances in which data is best fitted by our main models of interest (i.e. the self-reinforcing models *ConfUnspec*, *ConfSpec* and *Choice*) correspond most frequently to datasets that are also generated by these models.

To assess the quality of parameter recovery for the winning model *ConfUnspec*, we generated datasets (i.e. subjects) for which we systematically varied each model parameter with 250 equidistant values between sensible lower and upper bounds ($\alpha_r$: range 0.01–1; $\beta$: range 0.02–2; $\alpha_c$: range 0–1; $\gamma$: range 0–10). As in the model recovery analysis, experimental designs for each dataset were randomly sampled from the design generation function that was used in the behavioral experiment. Parameter recovery correlation matrices were constructed by correlating each varied generative parameter to all fitted parameters. To make the analysis robust against the specific settings of the respective other parameters, we performed this process for each node of the coarse parameter grid of size $k^N$ described above. For example, if the parameters of the coarse parameter grid were ($\alpha_r = 0.1$, $\beta = 0.4$, $\alpha_c = 0.1$, $\gamma = 1$), these exact parameters were used in the data generation for the construction of a correlation matrix, except for the parameter that was systematically varied for the estimation of a specific row in the correlation matrix.

S7 Fig shows that parameter recovery works well across various parameter regimes. Two edge cases deserve mention. First, we found that too low values for the confidence transfer parameter $\gamma$ impair the ability to recover the confidence learning rate $\alpha_c$ (if confidence has little effect on value estimates, the learning rate carries little weight either). Second, if choices become increasingly random (small values of $\beta$), recovery is likewise impaired to a certain degree. Nevertheless, for typical parameter values in our empirical data parameter recovery is sufficiently precise.

### Model generative performance

To better understand the qualitative behavior of the four best fitting models (*ConfUnspec*, *ConfSpec*, *Choice*, *Perseveration*) we simulated choice and confidence data across different

parameter settings. Specifically, we systematically varied parameters that influence the behavioral dynamics in phase 2, i.e. $\alpha_c$ (range 0–1 in steps of 0.5) and $\log \gamma$ (range 0–4 in 8 equidistant steps) for the *ConfUnspec* and *ConfSpec* model, $\lambda$ (range 0.5–10 in 7 exponential steps) for the *Choice* model and $\eta$ (range −1.5–1.5 in steps of 0.5) for the *Perseveration* model. Since we found that the effects of interest in phase 2 are not sensitive to the precise settings of $\alpha_r$ and $\beta$ (which shape behavior in phases with feedback), we fixed both parameters at 0.2, close to the average in the behavioral data and close to typical values in this context. For each model and parameter combination, 250 datasets were simulated.

Based on the simulated data, we computed four effects of interest: 1) *Performance effect*, defined as the average slope of proportion correct responses across the trials of phase 2; 2) *Confidence effect*, defined as the average slope of confidence across the trials of phase 2; 3) *Confidence x value effect*, defined as the slope for the interaction of CS-specific confidence slopes and corresponding CS values; 4) *Consistency effect*, defined as the increase of the proportion of consistent choices between the first and second occurrence of a choice pair versus the second and third occurrence of a choice pair.

## Supporting information

**S1 Appendix. Table A.** Mixed logistic regression on the dependent variable *correct* in phases 1 and 3. Performance increases significantly across phases with feedback (significant positive effect of *trial_number*). **Table B.** Mixed linear regression on the dependent variable *confidence* in phases 1 and 3. Confidence increases significantly across phases with feedback (significant positive effect of *trial_number*). **Table C.** Mixed logistic regression on the dependent variable *correct* in phase 2. Performance does not change significantly (non-significant effect of *trial_number*). **Table D.** Mixed linear regression on the dependent variable *confidence* in phase 2. Confidence increases significantly (significant positive effect of *trial_number*). **Table E.** Mixed logistic regression on the dependent variable *consistent* (coding whether a choice was consistent to the choice in the previous appearance of a CS pair) in phase 2. Consistency increases significantly with the number of appearances of a CS pair (significant positive effect of *trial_pair_repeat_nr*). **Table F.** Mixed linear regression on the dependent variable *rating_change* (subjective value rating post-phase-2 minus rating pre-phase-2). Ratings did not increase significantly with the objective value of the respective CS (no significant effect of *value*). **Table G.** Mixed linear regression on the dependent variable *rating_change* (subjective value rating post-phase-2 minus rating pre-phase-2). In comparison to the regression analysis in S6 Table, here we included the interaction of objective CS value (*value*) and the duration of phase 2 (*block_n-trials_phase2*).
(PDF)

**S1 Fig. Effect of value on rating changes (post-phase-2 minus pre-phase-2) in dependence of phase 2 duration.** Regression coefficient for the effect of value on rating changes across varying durations of phase 2.
(TIF)

**S2 Fig. Model evidence and N of free parameters.** Average Bayesian information criterion with s.e.m. across participants for all computational models considered and ordered by model fit. The number of parameters is displayed in parentheses. In line with the Akaike information criterion (see Fig 4 in the manuscript), *ConfUnspec* is the winning model.
(TIF)

**S3 Fig. Latent variables and behavioral predictions of the models *ConfSpec*, *Choice* and *Perseveration*.** See Fig 5 for details.
(TIF)

**S4 Fig. Generative model performance for four key effects in dependence of different parameter settings.** Model generative performance was assessed for the four best-performing models (*ConfUnspec*, *ConfSpec*, *Choice*, *Perseveration*). Model-based effects are shown as bar graphs, behavioral effects as dashed lines. *Performance effect*: model-based and behavioral graphs depict linear performance slopes in phase 2. *Confidence effect*: model-based and behavioral graphs depict linear confidence slopes in phase 2. *Confidence x value effect*: model-based and behavioral graphs depict slopes for the interaction of the Confidence effect and bandit value (cf. Fig 2C). *Consistency effect*: model-based and behavioral graphs depict the increase in choice consistency between the first and second occurrence of a choice pair versus the second and the third occurrence (cf. Fig 3A). Error bars for model-based effects indicate standard errors of the mean across 250 simulated subjects.
(TIF)

**S5 Fig. Model recovery (1): probability that a generative model *gen* is best fitted by a test model *fit*.** Rows represent generative models and each column within a row indicates the probability that a dataset was best fitted by a particular model. Note that the order of models is the same along both axes, but labels were omitted on the x-axis due to space constraints.
(TIF)

**S6 Fig. Model recovery (2): probability that a dataset best fitted by model *fit* was generated by model *gen*.** Rows represent the datasets in which the given model was best-fitting and each column within a row indicates the probability that the datasets were generated by a particular model. Note that the order of models is the same along both axes, but labels were omitted on the x-axis due to space constraints.
(TIF)

**S7 Fig. Parameter recovery.** Pearson correlation matrices between generative parameters and fitted parameters in dependence of different settings for $\beta$ and $\gamma$. The fixed $\beta$ and $\gamma$ values provided in the figure thus indicate the parameter values that were used for data generation in the construction of a recovery matrix. An exception is when $\beta$ and $\gamma$ were themselves varied–in these cases, the indicated values for $\beta$ and $\gamma$ do not apply; instead, different columns constitute internal replications for the recovery of $\beta$, and different rows constitute internal replications for the recovery of $\gamma$.
(TIF)

## Acknowledgments

We thank Yannick Schmidt for his assistance during the experiments. Computation has been performed on the HPC for Research cluster of the Berlin Institute of Health.

## Author Contributions

**Conceptualization:** Lena Esther Ptasczynski, Isa Steinecker, Philipp Sterzer, Matthias Guggenmos.

**Data curation:** Lena Esther Ptasczynski, Matthias Guggenmos.

**Formal analysis:** Lena Esther Ptasczynski, Matthias Guggenmos.

**Funding acquisition:** Philipp Sterzer, Matthias Guggenmos.

**Investigation:** Isa Steinecker, Matthias Guggenmos.

**Methodology:** Lena Esther Ptasczynski, Matthias Guggenmos.

**Project administration:** Matthias Guggenmos.

**Resources:** Philipp Sterzer, Matthias Guggenmos.

**Software:** Lena Esther Ptasczynski, Matthias Guggenmos.

**Supervision:** Philipp Sterzer, Matthias Guggenmos.

**Validation:** Lena Esther Ptasczynski, Matthias Guggenmos.

**Visualization:** Lena Esther Ptasczynski, Matthias Guggenmos.

**Writing – original draft:** Lena Esther Ptasczynski, Isa Steinecker, Philipp Sterzer, Matthias Guggenmos.

**Writing – review & editing:** Lena Esther Ptasczynski, Philipp Sterzer, Matthias Guggenmos.

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
