## [Decision Letter · Decision Letter 0]

7 Oct 2021

Dear Ptasczynski,

Thank you very much for submitting your manuscript "The value of confidence: Confidence prediction errors drive value-based learning in the absence of external feedback" for consideration at PLOS Computational Biology.

As with all papers reviewed by the journal, your manuscript was reviewed by members of the editorial board and by several independent reviewers. In light of the reviews (below this email), we would like to invite the resubmission of a significantly-revised version that takes into account the reviewers' comments.

Please note that we share all the major important concerns raised by the reviewers both at the conceptual level (why this effect?) and the technical level (model selection practice) and we expect them to be addressed in full (along with all the other points) in a thoroughly revised version of the manuscript that will be sent back to the original reviewers.

We cannot make any decision about publication until we have seen the revised manuscript and your response to the reviewers' comments. Your revised manuscript is also likely to be sent to reviewers for further evaluation.

Sincerely,

Stefano Palminteri

Associate Editor

PLOS Computational Biology

Samuel Gershman

Deputy Editor

PLOS Computational Biology

Reviewer's Responses to Questions

**Comments to the Authors:**

Reviewer #1: In this study, Ptasczynski and colleagues examine whether confidence can act as a learning signal, in the form of internal feedback, when external feedback is absent – an important but so far overlooked aspect of learning, presumably relevant to many real-life scenarios. They demonstrate that participants can maintain a running average of expected confidence, that is updated similarly as in standard RL mechanisms, and critically that is independent of the stimulus. They also rule out that stimuli are simply devalued in the absence of feedback. Overall, I found the rationale solid, and the paper is well written and can be an important contribution to the field.

The computational models are described comprehensively. Although it is definitely a strength that several computational models are included in the comparison + the best-fitting model variable trajectories are presented, we would like to see behavioral simulations of at least one alternative model, to validate the specificity of the mechanism at play.

Besides this main comment, I develop below by order of importance a number of comments to clarify some of the claims of the paper.

Main comment

It is interesting that the authors have included behavioral predictions alongside their model-based approach, but we need more to validate the winning model:

- Could the authors provide behavioral signature(s) such as how different the predictions would be for each or some of the five models at play? (guidance of Palminteri et al 2017 TICS and Wilson & Collins 2019 eLife). The authors have already fitted the models, and validate the winning model by comparing some of its dynamics (Fig 5) to those of participants (Fig 2). The key next step would be to similarly simulate the second-winning (at least) model to help us assess whether each of the models actually do make different qualitative predictions, and understand where the difference in AIC between models comes from.

- The winning model relies on AIC comparison. In Fig 4, is it average AIC across participants? because 4 points of AIC difference may not be much. Have the authors looked at e.g. BIC or model evidence too? Could the authors perform model recovery, to ensure that their fitting procedure actually can retrieve each of the models separately?

- Even for validation of the best-fitting model, comparing phase 2 in Fig 2B (participants) and Fig 5B (winning model) seems to reveal different qualitative features: the analyses indicate that confidence increases across phase2, but Fig 2B shows that confidence remains very flat; could the authors explain how we could reconcile this?

It seems important that the authors (i) can better isolate the winning model quantitatively or (ii) can show different qualitative predictions for the behavior of the model as compared to the other models or both (i) and (ii). This would make the claims stronger, or change the implications of the paper quite drastically (in case the winning model no longer wins).

General comments

Could the authors comment on the distribution of learning rates for confidence, that they are so low? Is that because of the confidence transfer parameter doing most of the job? Are these two parameters correlated? It would help to get a bit more insight on the mechanism here.

Regarding the assumption that confidence-related value updates only occur in the absence of external feedback, have the authors tried to extend this mechanism to the feedback case? That is, even if I receive a big reward, I might incorporate the related reward prediction error differently according to whether I was initially sure or unsure about my choice.

Could the authors comment on the fact that confidence-unspecific model may be appropriate for Phase2 which is limited in time and after which feedback resume, although in real-life situations where lack of feedback is more continuous, would they expect the confidence-unspecific mechanism to be less satisfactory, with a spreading of values across all items? The adaptiveness of this mechanism is commented on a bit in the Discussion, but would deserve more unpacking.

In the introduction, I was not sure how hypothesis #1 of increase of choice confidence is supported: is it the direct consequence of hypotheses #2 and #3 by which preferences become more marked and therefore choices easier? Or is there another mechanism at play here?

I did not understand what a positive effect of CS repetition number means; in what is this a hallmark of choice consistency? Doesn’t it mean that the same CS is selected within the CS pair? Relatedly, I don’t understand how Fig 3A is built; why do some consistency values not have a 2nd-3rd choice (orange) bar? Given the randomisation done, should it happen at least a bit?

Could the authors unpack their exploratory analysis : what is “value dependency”, and why does that show that longer phases without feedback have a “stronger effect of value on rating change”?

For choice devaluation (eq7), does it apply only to chosen stimuli, or both chosen and unchosen stimuli?

To provide evidence for a similarity between reward-learning and confidence-learning, it seems to me the most straightforward test would be to correlate learning rate reward and learning rate confidence parameters. This correlation is actually provided as a control analysis, but the focus is on correlation between learning rate reward and confidence transfer parameters. Could the authors explain this?

The initialisation of expected values and expected confidence are zero, but it seems to me that it would be reasonable that participants expect values to be in the middle, and confidence too, given that after some block they know that they could learn; do these initial values matter a lot of the trajectories of model variables?

Minor comments

The reward schedule seems clear from the description in Methods, although could the authors comment on a difference in performance across blocks in light of transfer effects? Could the authors confirm that the analyses focus on within-block comparison of the different phases, otherwise this might be problematic for across-block comparisons?

It would help in the text that the authors clarify what processes are the computational models trying to explain, particularly what aspects of the behavior (seeing Fig2B and 3B)?

I was unsure about the assumption that dissonance will be stronger for choices that are harder, was it a finding or a hypothesis? I did not understand to what extent the present proposal was consistent or inconsistent with the original Festinger idea. Could the authors unpack their reasoning?

In Fig 4 what are error bars? Are bars and error bars average and S.E.M. or S.D.?

Sometimes ‘variables’ is used where ‘parameters’ is meant, I believe

Why does the upper bound of softmax inf too?

Reviewer #2: In this manuscript, Ptasczynski and colleagues investigate behavioral and computational aspects of learning without feedback revealed by choices and confidence judgements. They used a simple instrumental-learning task which includes phases with and without external feedback and in which participants (n= 64) reported their confidence in addition to choices. The authors report signatures of self-reinforcement in phases without feedback, reflected in an increase of subjective confidence and choice consistency. They propose to account for these findings with a confidence-based learning models, which they compare with other potential models.

I found the manuscript generally well written, and appreciate the author care in providing an appropriate sample size (N = 64 – quite large for lab experiments) as well as a comprehensive analytical strategy.

Yet, and although I anticipate that the general topic, research question and approach would trigger some interest in the field, I feel that the current version of this manuscript falls short of providing a convincing demonstration of the effects and a compelling interpretation of the findings. I also have more conceptual interrogations about the general research question. I detail these concerns below.

Main concerns:

- First, I am having a hard time trying to figure out what exactly is (conceptually) this proposed self-reinforcement without feedback, in the context of this task. Is it an (normative) adaptation to some specific kind of ecological situations encountered in ecological settings ? Is this a bias ? Currently, it feels a bit like the research question is a bit of an ad-hoc effect, with an ad-hoc model. In other words: why should we expect confidence to rise in the absence of feedback ? Is there a limit to this increase ? The authors seem to frame this as a natural mechanism but I can imagine a lot of situations where the proposed mechanism would lead to terrible decision-making…

- I am also a bit unconvinced by the mere behavioral effects. Although the GLMM indicate a significant effect of trials on confidence in phase 2, the Figure 2B pictures an extremely small trend. Actually, it almost seems that confidence in the first trial of phase 3 is actually lower than (or at least no higher than) the confidence in the first trial of phase 2 – this would indicate that the apparent increase in confidence in the phase 2 is more a psychological effect of the absence of feedback than a self-reinforcement mechanism. Likewise, the effect on choice consistency are not very convincing: why limiting the analyses to trials 1-3 of phase 2 (Figure 3A)? One could actually compute a trial-by trial measure of choice repetition, and evaluate (and actually illustrate with a time-series/learning-curve figure) if/how choice repetition actually increases over the 20 trials of phase 2 – which should be the case under the authors’ hypothesis. This would correspond to the analysis reported in Table S5, which currently show a borderline effect (p = 0.032), but with some anomalies in the table (reported coefficient estimates are not included in the reported confidence intervals). Finally, and as transparently acknowledged by the authors (lines 151-160), there is no significantly detectable effect on value rating. So, overall, it seems that the model-free experimental evidence in favor of the self-reinforcement without feedback is at best tenuous. Of course, the data is the data (and I am not encouraging the authors to p-hack some new pattern of results here) – but in the absence of e.g. a strong replication, I will remain skeptic about the reality (and interpretation) of the behavioral effect.

- This allows me to transition to my second line of concerns – about the (model-based) interpretation of the effects. I am very puzzled by the winning, candidate model put forward by the authors. First, there are several aspects of the model rationale that I find questionable: e.g. the fact that there is confidence “transfer” only during phase 2 (one can imagine that this also happens when feedback is present), or that the choice probability (i.e. the softmax output) rather than the actual choice is used to determine the CS to which the confidence-value transfer is applied (lines 552-553). The author justifies it (lines 554-555) by explaining that using the actual choice would not make sense for the model, but it feels to me that this is not a good justification: one should adjust the model to the theory, rather than adjusting the theory to the constraints of the model. In connection to my general conceptual point, I also feel that the proposed model does not fully account for the observe behavioral patterns – this is maybe due to the fact that the authors only fit the model, and do not explore its generative performance (see (Palminteri et al., 2017)). For instance, if the model contributes to an evolution of option’s expected values (Figure 5A) shouldn’t it predict de facto a detectable increase in performance during phase 2 ?

- Given that the behavior actually falsifies the idea that the value (value ratings) change during phase 2, I have the feeling that a better (and more legitimate) model to explain the increase in choice consistency would be one leveraging a choice perseveration bias in the absence of feedback (possibly with the perseveration or choice temperature that is modulated by confidence) – see e.g. (Correa et al., 2018; Katahira, 2018; Rutledge et al., 2009).

- Finally, the model-based analysis does not really comply with current, state of the art modelling practices, and should include parameter recovery & model identification analyses (Wilson and Collins, 2019), as well as model simulations/falsifications (Palminteri et al., 2017).

Other concerns

- The use of AIC as the model-comparison criterion seems a arbitrary. This should be backed up, at the very least, by a model identification analysis (Wilson and Collins, 2019), or dropped for more principled model-comparison metrics. Likewise, the whole model-comparison exercise seems to consist in a random effect analysis of AIC (t-tests), which is not very standard nor principled. Some more principled solutions exists – see e.g. (Daunizeau et al., 2014).

- The initial values of expected values and confidence are set to 0 (544-545). Given that the task features only cues that have a positive expected value, shouldn’t v0 (which correspond to participants prior expectations) also be strictly positive (e.g. an average between the worst and best outcome) ? Likewise, it seems that initializing expected confidence at 0 does not really correspond to participants behavior/initial beliefs (Figure 2B).

- There are some issues with the GLMM results reported in Tables S3 and S5: the reported coefficients are not included in their respective CI.

- The authors report (lines 134-135) “By contrast, confidence increased across phase 2 (5.75 ± 0.04 [s.e.m.]; LMM: z = 3.12, p = 0.002; Figure 2B and Supplementary Table S4)”. I do not understand what the 5.75 ± 0.04 stands for.

References:

Correa, C.M.C., Noorman, S., Jiang, J., Palminteri, S., Cohen, M.X., Lebreton, M., and Gaal, S. van (2018). How the Level of Reward Awareness Changes the Computational and Electrophysiological Signatures of Reinforcement Learning. J. Neurosci. 38, 10338–10348.

Daunizeau, J., Adam, V., and Rigoux, L. (2014). VBA: A Probabilistic Treatment of Nonlinear Models for Neurobiological and Behavioural Data. PLOS Comput. Biol. 10, e1003441.

Katahira, K. (2018). The statistical structures of reinforcement learning with asymmetric value updates. J. Math. Psychol. 87, 31–45.

Palminteri, S., Wyart, V., and Koechlin, E. (2017). The Importance of Falsification in Computational Cognitive Modeling. Trends Cogn. Sci. 21, 425–433.

Rutledge, R.B., Lazzaro, S.C., Lau, B., Myers, C.E., Gluck, M.A., and Glimcher, P.W. (2009). Dopaminergic Drugs Modulate Learning Rates and Perseveration in Parkinson’s Patients in a Dynamic Foraging Task. J. Neurosci. 29, 15104–15114.

Wilson, R.C., and Collins, A.G. (2019). Ten simple rules for the computational modeling of behavioral data. ELife 8, e49547.

**Have the authors made all data and (if applicable) computational code underlying the findings in their manuscript fully available?**

Reviewer #1: Yes

Reviewer #2: None

PLOS authors have the option to publish the peer review history of their article (what does this mean?). If published, this will include your full peer review and any attached files.

Reviewer #1: No

Reviewer #2: No
---

## [Decision Letter · Decision Letter 1]

10 Feb 2022

Dear Dr Ptasczynski, 

Thank you very much for submitting your manuscript "The value of confidence: Confidence prediction errors drive value-based learning in the absence of external feedback" for consideration at PLOS Computational Biology.

As with all papers reviewed by the journal, your manuscript was reviewed by members of the editorial board and by several independent reviewers. In light of the reviews (below this email), we would like to invite the resubmission of a significantly-revised version that takes into account the reviewers' comments.

I agree with the reviewers’ assessment that the paper does not provide yet sufficient evidence in respect to generative performance, model recovery and parameter recovery. Some analyses have been added in this direction, but either they have not been implemented in the appropriate way or the results are (at least at this stage) inconclusive. We encourage you to carefully read the clear guidelines indicated by the reviewers in the previous and the current round of reviews on how to implement these analyses (and about what would be considered a positive outcome). Of note, additional information about these issues can be found in Wilson and Collins (eLife 2019, “ten simple rules” paper). Finally, if not done yet, once these analyses (generative, recoveries) are correctly implemented, I believe that sharing your code could help the evaluation of the revised manuscript.

We cannot make any decision about publication until we have seen the revised manuscript and your response to the reviewers' comments. Your revised manuscript is also likely to be sent to reviewers for further evaluation.

Sincerely,

Stefano Palminteri

Associate Editor

PLOS Computational Biology

Samuel Gershman

Deputy Editor

PLOS Computational Biology

Reviewer's Responses to Questions

**Comments to the Authors:**

Reviewer #1: I have now carefully read the responses to my comments for the study of Ptasczynski and colleagues, and I found that the manuscript has substantially improved.

My enthusiasm is a bit mitigated by the fact that the three main models do not present critically distinctive behavioral predictions, and, less importantly, they do not explain all aspects of behavioral data; although I understand that the study was not designed to arbitrate between these three models, that the data are what they are - and the text and claims of the paper are commensurate to the strength of the findings. Here are a few remaining points.

The authors discuss the adaptiveness of their identified self-reinforcement mechanism in terms of being potentially protective against memory leakage. In my understanding, this claim is mitigated by the three self-reinforcing model being consistent on most behavioral patterns. At the authors note, it is difficult to provide a functional interpretation for the adaptiveness of ConfUnspec because it might depend on the context. I appreciate that the authors’ model comparison indicates that BIC and AIC favour the ConfUnspec model, but still, in the absence of strong qualitative differences in behavior, it is difficult to appreciate where this differences in AIC/BIC come from. Instead, could the authors adapt and focus their discussion (lines 429-449) on the implication of the three self-reinforcement mechanisms compatible with the behavioral data, rather than of ConfUnspec specifically?

In response to Reviewer 2, I note that the parameter recovery is encouraging but incomplete. It is not sufficient to show that each of the 4 parameters can be captured, but that each parameter captures itself * and not the other * or * better than the other * parameters. From the correlations between fitted and simulated parameters in Fig S3, the authors can compute a form of 4x4 matrix, akin to the model recovery, but this time for parameter recovery. In other words, we currently see the correlations (simulated parameter i, fitted parameter i), and we need to see the correlations for all pairs (simulated parameter i, fitted parameter j).

Minor comments

In their Discussion, could the authors explicitly clarify the unique specificities of the present learning mechanism as compared to the learning mechanism proposed in Guggenmos et al 2016 eLife?

It would be useful to match the panels between e.g., Fig 5 and S4A, 2B and 5F to facilitate comparisons.

I find the notion of “more erratically update” ambiguous: could the authors replace with e.g. “with more variability”?

Reviewer #2: Ptasczynski revised their manuscript about confidence-driven learning in the absence of external feedback, to address the issues raised about their original submission by other reviewers and myself. First, I would like to thank the author for their attempt to address thoroughly and constructively the said issues: obviously, and lot of time and effort has been invested, and I find the revised manuscript much improved. Yet, in my view, some specific points of concern are still not satisfactorily addressed. I will try to re-state the analyses that I feel are still missing in the paper (or currently, mis-specified) and that would be needed to be added or corrected for me to recommend the manuscript for publication. All relates to the quality-control of the modelling.

1. Model generative performance. As far as I understand, the authors still not provide the simulation exercise required my myself and other reviewers, pertaining to model generative performance or model falsification (Palminteri et al., 2017). In response to these concerns the authors have produced graphs of the model latent variables / posterior predictive fits / fitted behavior. Although those are important per se, they do not address the question of model generative performance: those should rely on pure simulated behavior – and basically check that the model can generate the behavioral pattern of interest. What is the difference ? Simply, checking model simulated behavior rather than model fitted behavior. Why is this important ? Especially in RL, models tend to overfit behavior, due to the auto-correlation in the choice patterns. To understand this point, (Palminteri et al., 2017) makes a compelling point that e.g. Win-Stay Lose-Shift or choice-repetition models can fit learning behavior very well, but generally fail to produce the desired process of interest (e.g. reversal learning). Here because the proposed model really is a mechanism for the observed pattern of behavior, one need to be convinced that it can produce it, via pure simulations.

2. As for the falsification (which is again, currently problematically based on model fitting rather than model simulation), the authors currently show that all models predict the same confidence slope as a function of CS value. Can differential patterns be observed as a function of the other task parameters e.g. No trials in phase 1 ? Or interactions between these factors ? I feel that model falsification is a very desirable property, especially in a modelling paper that propose new mechanisms.

3. Model identification. Currently, I find the model identification analysis unsatisfactory. First, the metric is average AIC, which is hard to interpret, and can be driven by outliers : I suggest that the authors run several iterations of the modelling exercise, and report the probability of a model being identified as the best through their “preferred” model comparison exercise (p(fit|gen)). This probability is indispensable to infer what the model-comparison obtained from the actual data actually means. Second, it seems that the perseveration model wins all model comparison in the simulations. Actually, to deal with this case, I recommend that the author compute the reverse probability p(gen|fit), which would help the interpretation of the model comparison exercise – see e.g. Extended Data Fig. 5: in (Ciranka et al., 2022) for a similar approach/rationale.

4. Parameter recovery. The parameter recovery analyses currently does not give a fair representation of the ability of the modelling framework to correctly estimate parameters, because parameters are varied/estimated independently. This completely misses the fact that when fitting the data, multiple parameters can jointly compete to explain the same share of variance. For this reason, the parameter recovery should be ran by varying and fitting all parameters simultaneously.

References:

Ciranka, S., Linde-Domingo, J., Padezhki, I., Wicharz, C., Wu, C.M., and Spitzer, B. (2022). Asymmetric reinforcement learning facilitates human inference of transitive relations. Nat. Hum. Behav. 1–10.

Palminteri, S., Wyart, V., and Koechlin, E. (2017). The Importance of Falsification in Computational Cognitive Modeling. Trends Cogn. Sci. 21, 425–433.

**Have the authors made all data and (if applicable) computational code underlying the findings in their manuscript fully available?**

Reviewer #1: None

Reviewer #2: None

PLOS authors have the option to publish the peer review history of their article (what does this mean?). If published, this will include your full peer review and any attached files.

Reviewer #1: No

Reviewer #2: No
---

## [Decision Letter · Decision Letter 2]

28 Aug 2022

Dear Dr Ptasczynski

Thank you very much for submitting your manuscript "The value of confidence: Confidence prediction errors drive value-based learning in the absence of external feedback" for consideration at PLOS Computational Biology. As with all papers reviewed by the journal, your manuscript was reviewed by members of the editorial board and by several independent reviewers. The reviewers appreciated the attention to an important topic. Based on the reviews, we are likely to accept this manuscript for publication, providing that you modify the manuscript according to the review recommendations.

As you can see the authors appreciated the new efforts. However, it appears that some aspects of the presentation and description of the new analyses is lacking and does not allow a full understanding of your procedure and results. We ask to modify your manuscript to take into accounts this points before we can proceed with a formal acceptance.

Sincerely,

Stefano Palminteri

Academic Editor

PLOS Computational Biology

Samuel Gershman

Section Editor

PLOS Computational Biology

[LINK]

Reviewer's Responses to Questions

**Comments to the Authors:**

Reviewer #1: The authors have now provided additional analyses regarding the remaining points raised by other reviewers and myself about model validity, which has significantly improved the manuscript.

- Model recovery: I found the new Figure S4 hard to parse; we do not know what behavioral effects of participants are reproduced vs. not, it would be useful to indicate / report more precisely how human data compares.

- If I understood correctly, model simulations are not done from best-fitting parameters? It does not have to be independent from the fitted model parameters: quite the contrary, we want to know: within the regime of the best-fitting parameters, whether model simulations based on those parameters are able to reproduce the behavioral patterns of interest. But because the authors have swept across the whole landscape for selecting their parameter values, the relevant regimes are normally included.

- Falsification is not possible between the main candidate mechanisms, as acknowledged by the authors. As in my previous review, my enthusiasm remains mitigated by the fact that the main confidence-based models do not present critically distinctive behavioral predictions, and do not explain all aspects of behavioral data. However, I appreciate that the study was not designed to arbitrate between these mechanisms, and the authors have acknowledged this already, with the claims of the paper being commensurate to the strength of the findings.

Reviewer #2: After this second round of revision, I feel that the manuscript has further improved, and gives a fair account of the data and modelling exercise. I again commend the authors for their constructive involvement in this revision exercise. Nonetheless, and despite the fact that I usually try my best not to burden authors unnecessarily, I still feel that some sections need further clarification.

Especially, I am having a hard time understanding precisely what was done in the parameter recovery and model identification exercise:

Regarding parameter recovery, for instance, I don’t understand how the confusion matrices displayed in Supplementary Figure S7 can depict correlations between some parameters and beta or gamma, given that it seems that those should be fixed (by column or line, respectively). I’m also not sure to understand how exactly the parameters were varied (lines 680-683 and lines 709-714).

Regarding model identification, how was “the probability that datasets generated with a given model X were best fitted by a model Y” assessed ? Is it a frequency measure, based on a simple AIC comparison between models ? Or something else ?

Also, what exactly is referred to as a dataset in those cases ? A subject or a sample of subject equivalent to the study’s sample size ?

So I would like to encourage the author to comprehensively re-write these sections of the methods, taking the time (and space) to describe all steps and choices that have been made, and keeping in mind that reader should understand precisely what has been done, to the point of being able to reproduce the said control analyses.

**Have the authors made all data and (if applicable) computational code underlying the findings in their manuscript fully available?**

Reviewer #1: Yes

Reviewer #2: None

PLOS authors have the option to publish the peer review history of their article (what does this mean?). If published, this will include your full peer review and any attached files.

Reviewer #1: No

Reviewer #2: No

Figure Files:

Data Requirements:

Reproducibility:

References:

---

## [Editor Report · Decision Letter 3]

16 Sep 2022

Dear Dr Ptasczynski,

We are pleased to inform you that your manuscript 'The value of confidence: Confidence prediction errors drive value-based learning in the absence of external feedback' has been provisionally accepted for publication in PLOS Computational Biology.

Best regards,

Stefano Palminteri

Academic Editor

PLOS Computational Biology

Samuel Gershman

Section Editor

PLOS Computational Biology

---

## [Editor Report · Acceptance letter]

28 Sep 2022

PCOMPBIOL-D-21-01661R3 

The value of confidence: Confidence prediction errors drive value-based learning in the absence of external feedback

Dear Dr Ptasczynski,

I am pleased to inform you that your manuscript has been formally accepted for publication in PLOS Computational Biology. Your manuscript is now with our production department and you will be notified of the publication date in due course.

With kind regards,

Zsofia Freund
